# Enhanced Efficacy of PEGylated Liposomal Cisplatin: In Vitro and In Vivo Evaluation

**DOI:** 10.3390/ijms21020559

**Published:** 2020-01-15

**Authors:** Mohsen Ghaferi, Mohammad Javad Asadollahzadeh, Azim Akbarzadeh, Hasan Ebrahimi Shahmabadi, Seyed Ebrahim Alavi

**Affiliations:** 1Department of Chemical Engineering, Shahrood Branch, Islamic Azad University, Shahrood 36155-163, Iran; mn.ghaferi@gmail.com (M.G.); m.asadalahzadeh@gmail.com (M.J.A.); 2Department of Pilot Nanobiotechnology, Pasteur Institute of Iran, Tehran 1316943551, Iran; azimakbarzadeh1326@gmail.com; 3Department of Microbiology, School of Medicine, Rafsanjan University of Medical Sciences, Rafsanjan 7717933777, Iran; 4School of Pharmacy, The University of Queensland, Woolloongabba 4102, Australia

**Keywords:** cisplatin, drug delivery, nanoparticle, PEG, PEGylated liposome, polyethylene glycol, liposome

## Abstract

This study aims to evaluate the potency of cisplatin (Cispt)-loaded liposome (LCispt) and PEGylated liposome (PLCispt) as therapeutic nanoformulations in the treatment of bladder cancer (BC). Cispt was loaded into liposomes using reverse-phase evaporation method, and the formulations were characterized using dynamic light scattering, scanning electron microscopy, dialysis membrane, and Fourier-transform infrared spectroscopy (FTIR) methods. The results showed that the particles were formed in spherical monodispersed shapes with a nanoscale size (221–274 nm) and controlled drug release profile. The cytotoxicity effects of LCispt and PLCispt were assessed in an in vitro environment, and the results demonstrated that PLCispt caused a 2.4- and 1.9-fold increase in the cytotoxicity effects of Cispt after 24 and 48 h, respectively. The therapeutic and toxicity effects of the formulations were also assessed on BC-bearing rats. The results showed that PLCispt caused a 4.8-fold increase in the drug efficacy (tumor volume of 11 ± 0.5 and 2.3 ± 0.1 mm^3^ in Cispt and PLCispt receiver rats, respectively) and a 3.3-fold decrease in the toxicity effects of the drug (bodyweight gains of 3% and 10% in Cispt and PLCispt receiver rats, respectively). The results of toxicity were also confirmed by histopathological studies. Overall, this study suggests that the PEGylation of LCispt is a promising approach to achieve a nanoformulation with enhanced anticancer effects and reduced toxicity compared to Cispt for the treatment of BC.

## 1. Introduction

Bladder cancer (BC) is the most common urinary tract malignancy and the 7th leading cause of death from cancer (2.8% of all cancer deaths), with nearly 430,000 cases and 165,000 deaths annually worldwide [1]. Most cases of BC (75%) occur in men [1]. BC is a poor prognosis cancer with an unsatisfactory 5-year survival rate, owing to the difficulty in diagnosis and easy metastasis [2]. Chemotherapy is a conventional treatment of the disease [3].

Cisplatin (Cispt) is the most active cytotoxic agent for BC therapy [4]. It functions by forming a Cispt-DNA adduct and causing the induction of cell death through apoptosis [5]. In spite of therapeutic effects, Cispt has various severe side effects that restrict its clinical use, such as nephrotoxicity and neurotoxicity [6]. Nanomedicine and nanodelivery systems are relatively new scientific fields related to materials in the nanoscale range that are used as diagnostic devices or to deliver therapeutic agents to target sites in a controlled mode. Nanotechnology provides various opportunities for the treatment of chronic human diseases, such as cancer, through the site-specific and target-oriented delivery of correct therapeutics [7,8,9]. Kates et al. [10] synthesized Cispt-loaded Poly(L-aspartic acid sodium salt) (PAA) nanoparticles (140 ± 4 nm) and evaluated its efficacy on the treatment of BC in an in vivo environment. The results showed that BC-bearing rats treated with PAA-Cispt nanoparticles had no signs of T1 grade tumors, while 20% of rats treated with Cispt had T1 high-grade tumors in the bladder, indicating the role of PAA nanoparticles in increasing the therapeutic effects of Cispt in the treatment of BC. In another study, Sudha et al. [11] synthesized Cispt-loaded nano-diamino-tetrac (NDAT) particles (187 nm) and evaluated their efficacy in the treatment of BC in an in vivo environment. The results showed that the nanoparticles caused a significant reduction in the tumor volume compared to when Cispt was used (tumor volumes of 0.68 and 0.3 cm^3^ in Cispt and Cispt-loaded NDAT nanoparticles receiver animals, respectively). Liposome nanoparticles are another delivery system, which are widely used for the treatment of cancer in in vitro and in vivo environments [9,12,13].

Liposomes are nanocarriers that are compositionally identical to the cell membrane and are composed of phospholipids, which are self-enclosed to form lipid bilayers containing an aqueous core [14,15]. Phospholipids, as amphiphilic compounds, form polar shells in aqueous solutions. Due to the presence of an aqueous core and a lipid bilayer, liposomes are able to incorporate both hydrophilic and hydrophobic molecules [16]. They have various advantages, including non-toxicity, flexibility, biocompatibility, complete biodegradability, non-immunogenicity for systemic and non-systemic administration, the improvement of drug efficacy and therapeutic index, the improvement of drug stability, a decrease in drug toxicity, and ability to conjugate with site-specific ligands to achieve active targeting [17]. Liposomes are biodegradable and non-toxic materials, and therefore are biocompatible [18]. However, they suffer from low serum half-life [19]. This issue can be overcome through the incorporation of poly-(ethylene glycol) (PEG) into liposome composition to develop long-circulating liposomes with prolonged serum half-life [19]. PEGylation extends the circulation time of conjugated therapeutics through increasing their hydrophilicity and decreasing the glomerular filtration rate [19]. Moreover, the PEGylation of nanomaterials increases the tumor-targeting efficiency of chemotherapeutic drugs through enhanced permeation and retention effects [20].

This study aims to encapsulate Cispt into liposome and PEGylated liposome nanoparticles and evaluate their therapeutic and toxicity effects on the treatment of BC in in vitro and in vivo environments. For this purpose, the drug release behavior, in vitro cytotoxicity, and in vivo therapeutic and toxicity studies were performed using the dialysis membrane method, 3-[4,5-dimethylthiazole-2-yl]-2,5-diphenyltetrazolium bromide (MTT) assay, and histopathological studies, respectively.

## 2. Results and Discussion

### 2.1. Preparation of Nanoparticles

Cispt-loaded liposome (LCispt) and PEGylated liposome (PLCispt) were successfully synthesized by the reverse-phase evaporation method, which has been extensively used for liposome synthesis [21,22]. In this study, cholesterol was used to enhance membrane packing and reduce membrane fluidity [23]. Soybean lecithin was used as it has a lower amount of phosphatidylcholine compared to egg yolk lecithin, resulting in smaller vesicle size owing to the relatively larger dimension of the hydrophilic head group in the phospholipid molecule [24]. Researchers in one study [25] showed that liposomes prepared with soybean lecithin were the most uniform in terms of structure and showed higher stability compared to liposomes prepared from egg yolk lecithin.

In the present study, PEG was used as a non-toxic polymer [26]. It is used for different purposes due to its aqueous solubility, low immunogenicity, and antigenicity and proper excretion kinetics [26,27]. PEGylation of liposomes can form a more rigid liposomal layer to inhibit the coalescence of particles. The rigidity is a critical factor that influences the delivery efficiency of bioactive compounds, such as the storage stability and release profile [24].

### 2.2. Characterization of Nanoparticles

The size of nanoparticles is a critical factor for determining the efficiency of loaded chemotherapeutics, in that smaller nanoparticles have higher cellular uptake and cellular transfection efficiency. This leads to an increase in the intracellular concentration of the loaded chemotherapeutics [28]. Nanoparticles with a size less than 300 nm can impressively penetrate target cells and exert their pharmaceutical effects [29]. Also, size distribution is an important factor that influences the stability of colloidal dispersion. A liposomal population should be as homogenous as possible to inhibit unwanted coalescence [30].

The results of the present study demonstrate that homogenous nanoparticles with a negative zeta potential and sizes ranging from 221 to 274 nm were formed (Table 1). Also, the loading of Cispt into liposome caused an increase in the size, indicating the encapsulation of the drug into nanoparticles [31]. In addition, the results show that PEGylation caused a decrease in the size of nanoparticles. As PEG has a hydrophilic nature and high permeability, it penetrates liposomal layers and presses them together [21,32]. The size distribution for free liposome was significantly higher compared to LCispt and PLCispt formulations, which could have resulted from the effects of Cispt as a positive charge molecule in increasing the nanoparticles’ homogeneity.

All nanoformulations had negative zeta potentials, resulting in them having proper stability in aqueous solutions with low ionic strength. This finding results from the fact that particles with the same charge (negative or positive) repulse each other [33], and this prevents aggregation. However, the zeta potential of LCispt was more positive than that of liposome due to the positive charge of Cispt [33]. Also, PEGylation caused an increase in the zeta potentials of liposomes due to the surface coating of nanoparticles by PEG chains [34,35].

The morphology of the nanoparticles was evaluated using light microscopy and the results showed that spherical unilamellar vesicles were formed without aggregation (Figure 1).

The nanoparticles were assessed using a scanning electron microscopy (SEM) instrument. The results showed that monodispersed nanoparticles with smooth surfaces were formed without surface fractures or pitting (Figure 2).

Regarding therapeutic applications using drug delivery systems, the highest amount of drug loading efficiency (LE) must be achieved to minimize the risk of carrier damage caused by hydrolysis. Thus, LE is an important factor in the development of a successful drug delivery system [36].

In this study, the drug encapsulation efficiency (EE) and LE values of LCispt were 34 ± 1.6 and 9.9 ± 0.49%, respectively, while these amounts for PLCispt were 37 ± 1.8 and 10.7 ± 0.52%, respectively. According to the results, PEGylation caused an increase in EE and LE. PEG enhances drug solubility, and as a result promotes drug LE. The relatively low amounts of EE and LE resulted from the poor water solubility and low lipophilicity of Cispt [37].

Maintaining drugs’ chemical structures is critical because changes in their chemical structure can cause changes in their biological activity [38]. In the present study, the Fourier-transform infrared spectroscopy (FTIR) method was used to evaluate the chemical structure of Cispt. The results showed that: (i) a strong peak related to Pt-NH_3_ was obtained at 463 cm^−1^; (ii) two peaks related to Pt-Cl bond were provided at 338 cm^−1^ and 362 cm^−1^; and (iii) a series peaks related to NH_3_ was provided at 1290 cm^−1^, 1550 cm^−1^, 1680 cm^−1^, 3300 cm^−1^, and 3500 cm^−1^ [39,40]. As Figure 3 shows, the chemical bonds of Cispt were preserved after being loaded into liposomes (i.e., Cispt was physically loaded into the liposomes).

### 2.3. Drug Release Study

In the drug release process, a device or a composite release a drug molecule in a controlled way. The drug is then subjected to absorption, distribution, metabolism, and excretion, and eventually will become available for pharmaceutical function [41]. To achieve and preserve the effective therapeutic concentration of a drug in plasma several doses are needed daily, which may result in considerable fluctuations in the plasma drug concentration [42]. These fluctuations can lead to a fall in the drug concentration in plasma below the minimum effective concentration or an increase beyond the minimum toxic concentration, resulting in the lack of therapeutic benefit or undesirable toxic effects [43]. Sustained-release and controlled-release drug delivery systems are able to decrease these undesired fluctuations leading to a decrease in the side effects, while the therapeutic outcome of the drug is improved [41].

In the present study, the cumulative drug release from Cispt solution, LCispt, and PLCispt was measured using the dialysis membrane method. As Figure 4 shows, Cispt was released from the Cispt solution in a burst-release manner, in which 56 ± 2.8% of Cispt was released in the first hour of the study, after which time the trend slowed. However, 92 ± 4.5% of the total drug was released over 8 h. Also, the drug release for LCispt and PLCispt was initiated with a burst release, in which 23 ± 1.1 and 20.5 ± 1% of the total drug amounts released for LCispt and PLCispt occurred in the first hour of the study. The drug release for both nanoformulations was continued with a mild increasing trend and in a controlled manner, for which 43 ± 2 and 39 ± 1.9% of the loaded drug amounts were released after 48 h for LCispt and PLCispt, respectively. As the results show (Figure 4), a lower amount (4%) of loaded Cispt was released from PLCispt compared to LCispt. This finding results from the fact that PEG functions as a cover and reduces drug leakage from the carrier, increasing drug stability [32]. Our results were in agreement with the results of Kuang et al. study [44], where 32.7% and 29.1% of Cispt were released from non-PEGylated and PEGylated liposomal Cispt, respectively, after 36 h. Overall, the low amounts of drug release from liposomes in the present study reflected the prolonged-release property of these carriers, especially for the PLCispt.

### 2.4. Cytotoxicity Study

Nanoparticles are able to enhance the therapeutic efficacy of anticancer drugs [7], owing to their ability to increase the drug concentration in tumor cells through enhancing the drug circulation time. Also, nanoparticles have the capability of delivering drugs to their target sites in an in vivo environment, and thereby they can inhibit the drugs’ side effects on normal cells [6,45,46].

In the current study, the cytotoxicity effects of LCispt and PLCispt on HTB-9 cells in an in vitro environment were assessed using MTT assay and the half-maximal inhibitory concentration (IC_50_) was measured using GraphPad Prism. As Figure 5 shows, the encapsulation of Cispt into liposomes caused a significant increase (*p* < 0.05) in the cytotoxicity effects of the drug, which were more prominent after the first 24 h (2.2-and 1.8-fold after 24 and 48 h incubation, respectively). Frangos et al. [47] showed that the encapsulation of INF-α into liposomes led to an increase in the cytotoxicity effects of INF-α against BC cell line 253J by more than 2.1-fold. In the current study, PLCispt demonstrated insignificant increases in cytotoxicity compared to LCispt by 5% and 6% after 24 and 48 h incubation, respectively. This low enhancement in the cytotoxicity effects of PLCispt compared to LCispt might be related to the drug release profiles of both formulations, in which the amount of released Cispt from PLCispt was slightly lower than LCispt. The cytotoxicity effects of PLCispt compared to Cispt were increased by 2.4- and 1.9-fold after 24 and 48 h incubation, respectively. This finding results from the fact that PEG improves drug stability and reduces drug release, leading to an increase in drug cytotoxicity effects [32]. In addition, in the present study, it was found that the cytotoxicity effects of all formulations increased by increasing the incubation time (Figure 5).

### 2.5. Stability Study

Avoiding premature drug release from drug delivery systems is important to reduce drug toxicity [48]. Nanocarriers play a critical role in the development of formulations as they increase stability [49]. In this study, to evaluate the stability of PLCispt, the cytotoxicity effects of PLCispt were assessed two months after its synthesis and compared to the results from the production day. As Figure 6 shows, the cytotoxicity effects of PLCispt were not significantly changed (*p* > 0.05), and the formulation preserved the stability and cytotoxicity effects of Cispt (IC_50_ of 16 ± 0.8 and 15 ± 0.7 µM after 24 and 48 h incubation, respectively, at the production time; and IC_50_ of 16.5 ± 0.8 and 15.3 ± 0.7 µM from 24 and 48 h incubation, respectively, after 2 months). Therefore, PEGylated liposomes can be considered as a competent carrier to preserve Cispt stability, and as a result, its anticancer effect [39]. Also, the size, size distribution, and zeta potential of PLCispt were measured two months after its preparation, and the results showed that these values were not significantly (*p* > 0.05) changed compared to those results evaluated at the production time (Table 2). These results were in agreement with the results for cytotoxicity, indicating the high stability of PLCispt formulation. The PEGylation of liposome nanoparticles caused an increase in the stability of the particles.

### 2.6. In Vivo Antitumor Efficacy of the Formulations

In the current study, an orthotopic model of BC was successfully developed using BBN as a bladder carcinogen [50] in female Wistar rats according to the previous studies [51,52]. Animals were administered N-Butyl-N-(4-hydroxybutyl)-nitrosamine (BBN) (0.05% in drinking water) for 8 weeks and then intraperitoneally received no treatment (as the control group), phosphate-buffered saline (PBS) (as a vehicle control group), Cispt, LCispt, and PLCispt, respectively. Their weights were measured weekly until the end of the study (week 20). At the end of week 20, the animals were anesthetized and sacrificed to remove the urinary bladder to evaluate the lesions. Tumors were developed in all rats receiving BBN, and all animals completed the 20 week protocol with no mortality. The tumor volumes were then measured to be 27 ± 1.3, 26 ± 1.3, 11 ± 0.5, 6 ± 0.3, and 2.3 ± 0.1 mm^3^ in the control, PBS, Cispt, LCispt, and PLCispt groups, respectively (Figure 7). The results showed that tumor volume was decreased by 2.4- to 11.7-fold in all groups that received Cispt compared to the control group. Liposome caused a further increase in the Cispt efficacy (*p* < 0.05) and the reduction of the tumor volume by 1.8-fold in LCispt and 4.8-fold in PLCispt. These results were in agreement with the results of Miyazaki et al. [53], where *Mycobacterium bovis bacillus* Calmette Guérin cell wall (BCG-CW)-loaded liposomes caused 1.6-fold reduction in the tumor volume compared to BCG-CW (2.1 ± 0.3 and 3.3 ± 0.4 mm^3^) in BC-bearing rats. In the present study, PLCispt was found to be more efficacious compared to LCispt in increasing the Cispt efficacy and reduction of tumor volume, as it caused 2.6-fold reduction in the tumor volume compared to LCispt, demonstrating the role of PEGylation in enhancing the potency of liposome nanoparticles after intraperitoneal administration. In this regard, the results of one study [54] showed that PEGylated liposomes could be detected in the blood 30 h after intraperitoneal injection, while this value for non-PEGylated liposome was 7 h. Therefore, coating the surfaces of liposomes with PEG improves carrier stability in blood circulation and extends the drug release [31], and as a result enhances the chance of drug delivery to the target site. Also, the results of the present study showed that the tumor number was significantly decreased in tumor-bearing rats receiving PLCispt compared to other groups (the tumor numbers in control, PBS, Cispt, LCispt, and PLCispt groups were 21.8 ± 1.1, 20 ± 0.9, 12.1 ± 0.6, 8.1 ± 0.4, and 5.3 ± 0.2, respectively). Moreover, tumor growth inhibition index (TGII) was measured to confirm the anticancer effects of the formulations [33], which was 4%, 59%, 78%, and 91% in PBS, Cispt, LCispt, and PLCispt groups, respectively, confirming the higher anticancer effects of PLCispt compared to the other formulations.

To evaluate the toxicity effects of the formulations, changes in the bodyweight of animals and in the serum concentrations of kidney- and liver-related factors were measured, including blood urea nitrogen (BUN), creatinine, aspartate transaminase (AST), alanine transaminase (ALT), and alkaline phosphatase (ALP) [55]; and histopathological studies were performed. For this purpose, the bodyweight changes were recorded weekly after initiating the treatment. As Figure 8 shows, all tumor-bearing animals receiving Cispt demonstrated lower bodyweight gain (BWG) compared to the control group (no treatment group), especially two weeks after initiating the treatment. At the end of week 20, the BWG values in the control, Cispt, LCispt, and PLCispt groups were 11%, 3%, 8%, and 10%, respectively, confirming the potency of PLCispt to decrease the toxicity effects of Cispt by 3.3-fold (Figure 8). The potency of liposome to reduce the Cispt toxicity and increase the BWG index has also been reported before [56]. The toxicity effects of the formulations were also evaluated using the measurement of the serum concentrations of BUN, creatinine, AST, ALT, and ALP, and the results showed that LCispt and PLCispt caused a significant decrease (*p* < 0.05) in the serum concentration of these factors (Table 3), indicating the role of liposome in decreasing the toxicity of Cispt. However, PLCispt caused a further reduction (*p* < 0.05) in serum levels of these factors compared to LCispt, indicating the role of PEGylation in decreasing Cispt toxicity. This finding might result from the fact that PEGylation caused an increase in liposome stability and also a decrease in drug release compared to LCispt, resulting in toxicity reduction. Also, the toxicity of the formulations was evaluated using histopathological studies, and the results were inconsistent with the results of the body-weight changes, in that LCispt and PLCispt obtained the lower scores compared to Cispt (Table 3 and Figure 9), demonstrating a decrease in histopathological lesions. The potency of liposome to decrease the histopathological effects of Cispt has also been reported previously [57], where rats receiving intraperitoneally Cispt (1 mg/kg, 15 times) showed moderate (3 of 5) and severe (2 of 5) acute tubular necrosis, while rats administered LCispt (1 mg/kg, 15 times) showed no evidence of this disorder. Overall, the results showed that PEGylated liposome is a promising carrier for improving the therapeutic effects of Cispt and decreasing its toxicity effects.

## 3. Materials and Methods

### 3.1. Materials

Hydrogen chloride, MTT, PBS, soybean lecithin, cholesterol, sodium hydroxide, mannitol, H&E, dialysis bag (cut-off of 10,000 Da), ethanol (EtOH), xylazine, ketamine hydrochloride, and Cispt were purchased from Sigma-Aldrich (St. Louis, MO, USA). Roswell Park Memorial Institute (RPMI)-1640 medium, penicillin and streptomycin antibiotics, and fetal bovine serum (FBS) were purchased from Gibco (Waltham, MA, USA). BBN was purchased from J&K Scientific Co., Ltd. (Beijing, China). PEG2000 was purchased from Kimiagaran Emrooz Chemical Ind. (Arak, Iran). Female Wistar rats (8 weeks, 250 g) and bladder carcinoma cell line (HTB-9) were supplied by Pasteur Institute of Iran, Tehran. All other materials were of analytical grade. Deionized water was used throughout the study.

### 3.2. Preparation of Nanoparticles

LCispt and PLCispt were prepared using the reverse-phase evaporation method. To prepare LCispt nanoparticles, 120 mg of lecithin, 45 mg of cholesterol (at the molar ratio of 58:42), and 20 mg of Cispt were added into 200 mL of 98% EtOH and stirred (200 rpm, 1 h, room temperature). Next, EtOH was evaporated using rotary evaporator (Heidolph Co., Schwabach, Germany) to produce a thin yellow layer. The resulting mixture was then added into 10 mL of PBS and stirred (200 rpm, 4 h, room temperature). To prepare PLCispt, 27 mg PEG2000 was added into the mixture (lecithin, cholesterol, and PEG at the molar ratio of 55:40:5). Both formulations were sonicated for 30 min (Bandelin Sonorex Digitec, Berlin, Germany; 60 Hz) to manufacture particles.

### 3.3. Nanoparticles Characterization

To characterize LCispt and PLCispt in terms of size, size distribution, and zeta potential, a Zetasizer instrument (ZEN 3600, Malvern Instruments Ltd., Worcestershire, UK) was used. For this purpose, 0.5 mg/mL of both formulations was prepared and introduced to the instrument.

Also, EE and LE were determined using high-resolution continuum source graphite furnace atomic absorption spectrometry (HR-CS GFAAS) model ContrAA 700 (Analytik Jena, AG, Jena, Germany) using H2PtCl6 (Sigma Aldrich, St. Louis, MO, USA) as a standard. For this purpose, 5 mg/mL of both formulations was precipitated (15,000 rpm, 30 min, 4 °C) to obtain supernatants. The concentration of Cispt in the supernatants was determined using AAS at the absorbance of 265 nm. EE (%) and LE (%) were then calculated using the formulae below:(1)$$\mathrm{EE}\;(\%)\;=\;\frac{\mathrm{Encapsulated}\;\mathrm{drug}\;(\mathrm{mg})}{\mathrm{Total}\;\mathrm{drug}\;(\mathrm{mg})}\;\times \;100$$
(2)$$\mathrm{LE}\;(\%)\;=\;\frac{\mathrm{Initial}\;\mathrm{drug}\;\mathrm{concentration}\;(\mathrm{mg})\;-\;\mathrm{drug}\;\mathrm{concentration}\;\mathrm{in}\;\mathrm{supernatant}\;(\mathrm{mg})}{\mathrm{Initial}\;\mathrm{drug}\;\mathrm{concentration}\;(\mathrm{mg})}\;\times \;100$$

Both formulations were observed under SEM using a model XL30 Philips SEM (Eindhoven, the Netherlands) and light microscopy (Nikon Eclipse E200; Nikon, Tokyo, Japan). For this purpose, 1 mL of mannitol (3.0% *w/v*) was individually added into 1 mL of each nanoformulation and lyophilized (Edwards High Vacuum, Manor Royal, Crawley, Sussex, UK). The resulting powders were then visualized after gold metallization by the SEM instrument.

Also, to determine whether the chemical structure of Cispt was preserved after loading into liposomes, the FTIR spectroscopy technique was used. For this purpose, the suspension of PLCispt was centrifuged (15,000 rpm, 30 min, 4 °C) and the precipitate was left at room temperature to dry. Then, a mixture of the dried pellet (2 mg) and bromide potassium (200 mg) was prepared and pressed. The obtained tablet was then analysed using the FTIR instrument (Nicolet 740SX FTIR spectrophotometer with an mercury cadmium telluride (MCT)-B (wide band MCT – 400 cm^-1^ cut-off) detector (Madison, WI, USA)).

### 3.4. Drug Release Study

The dialysis membrane technique was used to study drug release. For this purpose, 2 mL of LCispt and PLCispt (1.48 and 1.36 mg of Cispt, respectively) was centrifuged (15,000 rpm, 30 min, 4 °C), and the supernatants were discarded to remove the free drug. The pellets were resuspended in 5 mL of PBS and transferred into dialysis bags (cut-off limit of 10 kDa). Additionally, a solution of standard Cispt (5 mL, 0.3 mg/mL) was prepared in PBS and transferred into another dialysis bag. The bags were then immersed in 100 mL of PBS and stirred (200 rpm, room temperature). At different time points (0, 1, 2, 3, 8, 12, 18, 24, 30, 36, 42, and 48 h), 1 mL of PBS was collected to determine Cispt concentration using AAS method and replaced with 1 mL of fresh PBS. Cumulative drug release percentage was calculated using the formula below:(3)$$\mathrm{Drug}\;\mathrm{release}\;(\%)\;=\;\frac{\mathrm{Released}\;\mathrm{drug}\;\mathrm{from}\;\mathrm{particles}\;(\mathrm{mg})}{\mathrm{Total}\;\mathrm{drug}\;\mathrm{in}\;\mathrm{particles}\;(\mathrm{mg})}\;\times \;100$$

### 3.5. Cytotoxicity Study

The cytotoxicity effects of LCispt and PLCispt compared to the standard Cispt were assessed using an MTT assay on HTB-9 cells. For this purpose, HTB-9 cells were seeded in RPMI-1640 medium supplemented with 10% FBS and 1% penicillin or streptomycin antibiotics in a 96-well plate at the density of 10^4^ cells/well, and incubated for 24 h (5% CO_2_, 37 °C). The media were discarded, and the cells were treated with LCispt and PLCispt at the drug concentrations of 2, 4, 8, 16, 32, 64, 128, and 256 µM. After 24, 48, and 72 h incubation (5% CO_2_, 37 °C), the media were replaced with 100 µL of MTT (0.5 mg/mL PBS) and the cells were incubated (5% CO_2_, 37 °C) for 4 h. Then, MTT was discarded, and isopropanol was added to dissolve the formazan crystals. The absorbance was read at 540 nm using a microplate scanning spectrophotometer (ELISA reader, Organon Teknika, Boxtel, the Netherlands) and the cell viability was calculated using the following formula.
(4)$$\mathrm{Cell}\;\mathrm{viability}\;(\%)\;=\;\frac{{\mathrm{Absorbance}}_{sample}{\;-\;\mathrm{Absorbance}}_{background}}{{\mathrm{Absorbance}}_{negativecontrol}{\;-\;\mathrm{Absorbance}}_{background}}\;\times \;100$$

Only media and the cells treated with media only were considered as background and negative control, respectively. IC_50_ values of the standard Cispt, LCispt, and PLCispt were calculated using GraphPad Prism software version 8.00. All experiments were performed in triplicate.

### 3.6. Stability Study

Two months after the preparation of PLCispt (storage at 24 ± 1 °C and 55 ± 5% humidity), the cytotoxicity effects of the formulation compared to those obtained at the production time were determined using an MTT assay on HTB-9 cell line, as previously described. Additionally, the size, size distribution, and zeta potential values of the nanoformulation were measured.

### 3.7. In Vivo Antitumor Efficacy of the Formulations

Fifty female Wistar rats (age: 8 weeks; weight: 250 g) were housed (25 ± 2 °C, 12 h light/12 h dark cycle, 55 ± 5% humidity) with free access to standard food and water. The experiments were approved by the Animal Experimentation Ethics Committee of Pasteur Institute of Iran, Tehran (No# IR.PII.REC.1395.19;27 February 2017). After two weeks, all rats were received 0.05% BBN in drinking water for 8 weeks to develop the BC. The rats were then divided into five groups and received no treatment (as a control group), PBS (as vehicle group), Cispt, LCispt, and PLCispt, respectively. The animals received the treatment (1.5 mg/kg Cispt net content) intraperitoneally every 72 h for 6 cycles. After 20 weeks, the animals were anesthetized (50 mg/kg xylazine and ketamine anesthesia), heart blood samples were obtained, and the animals were sacrificed by cervical dislocation. The urinary bladders of rats were immediately removed and examined for grossly visible lesions. Additionally, the liver and kidneys were immediately removed and placed in formaldehyde for further analysis. The number of tumors in each rat and the volume of each tumor were measured to calculate the tumor incidence in each group and the mean tumor volume in each rat. A tumor was considered as a lesion of > 0.5 mm in diameter. Also, tumor volume (mm^3^) was measured by the formula below:(5)$$\mathrm{Tumor}\;\mathrm{volume}\;=\;\left({\mathrm{Length}\;\times \;\mathrm{Width}}^{2}\right)\;\times \;0.5$$

TGII was also determined using the formula below:(6)$$\mathrm{TGII}=\frac{\mathrm{Mean}\;\mathrm{tumor}\;\mathrm{weight}\;\mathrm{in}\;\mathrm{control}\;\mathrm{group}\;-\;\mathrm{Mean}\;\mathrm{tumor}\;\mathrm{weight}\;\mathrm{in}\;\mathrm{other}\;\mathrm{groups}}{\mathrm{Mean}\;\mathrm{tumor}\;\mathrm{weight}\;\mathrm{in}\;\mathrm{control}\;\mathrm{group}}\;\times \;100\%$$

The rats’ weight changes were also determined to evaluate the toxicity of the formulations. In addition, toxicity was evaluated by measuring the serum concentrations of BUN, creatinine, AST, ALT, and ALP using the related measurement kits (Pars Azmoon Company, Tehran, Iran) and according to the manufacturer’s instructions.

### 3.8. Histological Evaluation

To evaluate histological changes in kidney and liver, successive sections of paraffin-embedded tissue were prepared and stained with H&E. The organ toxicity was scored as no side effect (0), low-grade side effect (1), and high-grade side effect (2).

### 3.9. Statistical Analysis

Statistical analyses were performed using GraphPad Prism software version 8.00, and statistical differences were analyzed by one-way analysis of variance (ANOVA) test.

## 4. Conclusions

In this study, Cispt was loaded into liposome and PEGylated liposome particles using the reverse-phase evaporation method. The results showed that the particles were synthesized at the nanoscale and released Cispt in a controlled manner. The cytotoxicity effects of Cispt were increased by 2.4- and 1.9-fold after 24 and 48 h, respectively, when PLCispt was used. PLCispt was also found to maintain the cytotoxicity effects of the drug two months after the preparation time. To evaluate the therapeutic and toxicity effects of the formulations, an in vivo model of BC was induced in the female Wistar rats, and the results showed that PLCispt caused a significant decrease in the tumor volume (by 4.8-fold) and simultaneously a 3.3-fold reduction in the toxicity effects compared to Cispt. Overall, the in vitro and in vivo results demonstrated that PEGylation of the liposomal Cispt is a promising approach to obtain nanoformulations with enhanced anticancer effects and reduced toxicity compared to LCispt for the treatment of BC.

## Figures and Tables

**Figure 1 ijms-21-00559-f001:**
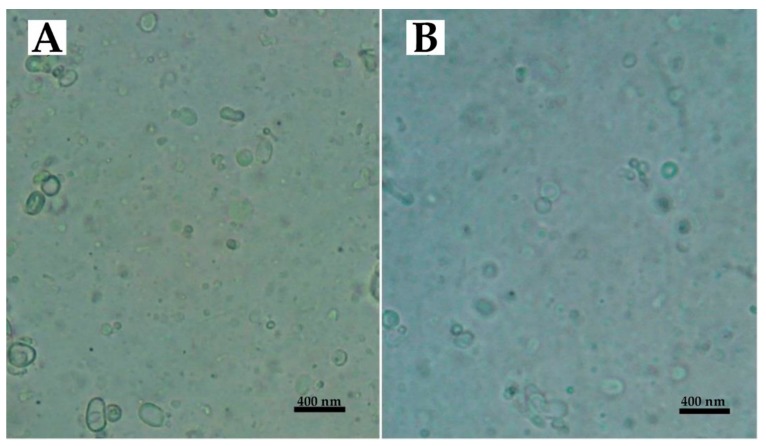
Light microscopy of (**A**) LCispt and (**B**) PLCispt.

**Figure 2 ijms-21-00559-f002:**
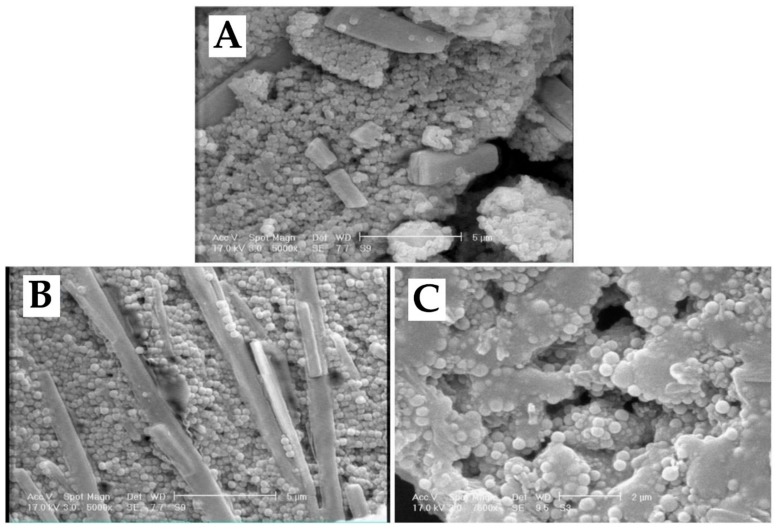
Scanning electron microscopy (SEM) images of (**A**) liposome, (**B**) LCispt, and (**C**) PLCispt prepared by the reverse-phase evaporation method.

**Figure 3 ijms-21-00559-f003:**
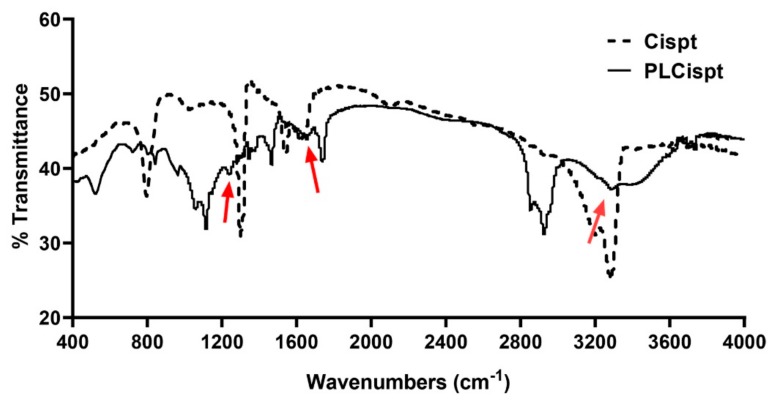
Fourier-transform infrared spectroscopy (FTIR) spectrum of the PLCispt. Arrows show the position of chemical bonds of Cispt which are preserved in PLCispt.

**Figure 4 ijms-21-00559-f004:**
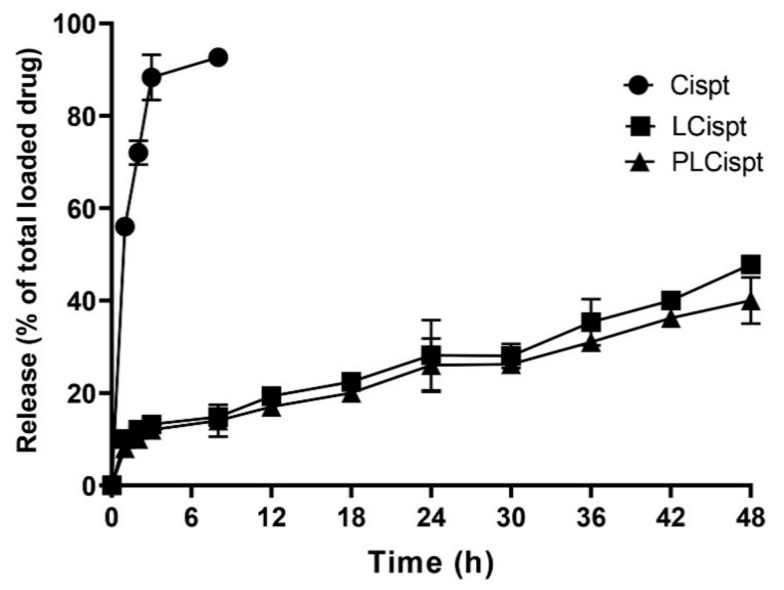
Release pattern of Cispt from Cispt, LCispt, and PLCispt. Statistical analyses were performed using one-way analysis of variance (ANOVA) and F-tests. The data are expressed as mean ± SD (*n* = 3).

**Figure 5 ijms-21-00559-f005:**
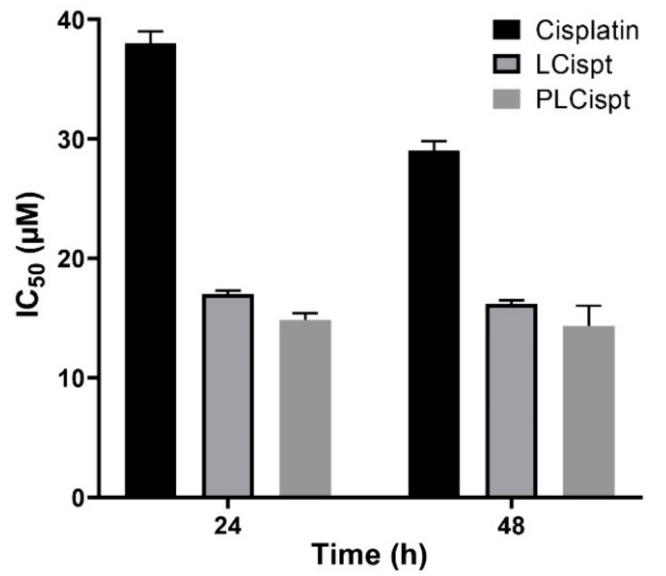
Cytotoxicity effects (IC_50_) of Cispt, LCispt, and PLCispt after 24 and 48 h incubation. The data are expressed as mean ± SD (*n* = 3).

**Figure 6 ijms-21-00559-f006:**
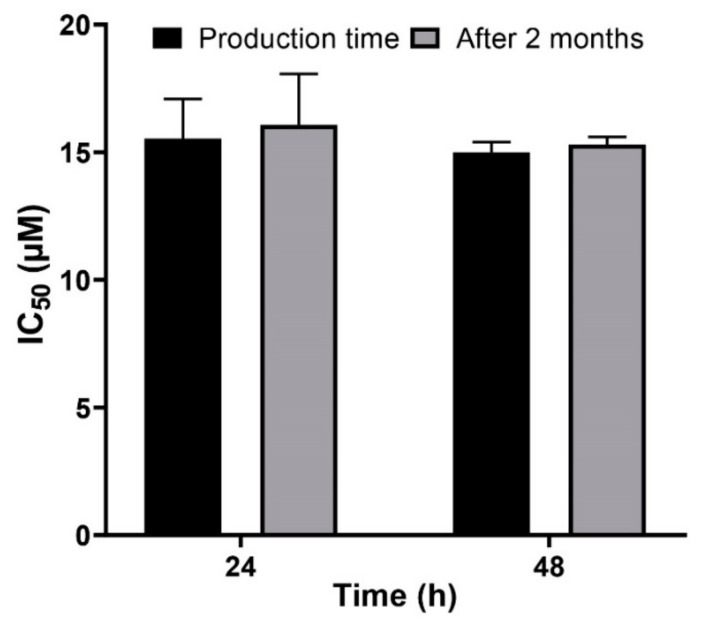
Evaluation of the cytotoxicity effects of PLCispt at the production time compared to two months after the synthesis. The data are expressed as mean ± SD (*n* = 3).

**Figure 7 ijms-21-00559-f007:**
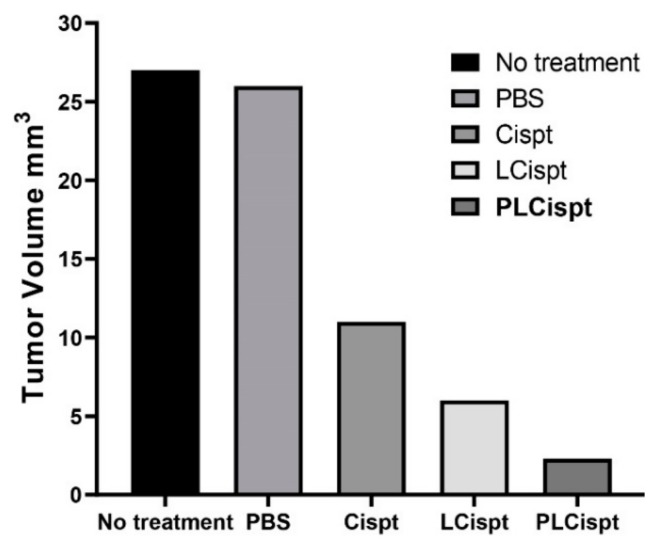
The tumor volume (mm^3^) in bladder cancer (BC)-bearing rats that received different formulations, including phosphate-buffered saline (PBS), Cispt, LCispt, and PLCispt.

**Figure 8 ijms-21-00559-f008:**
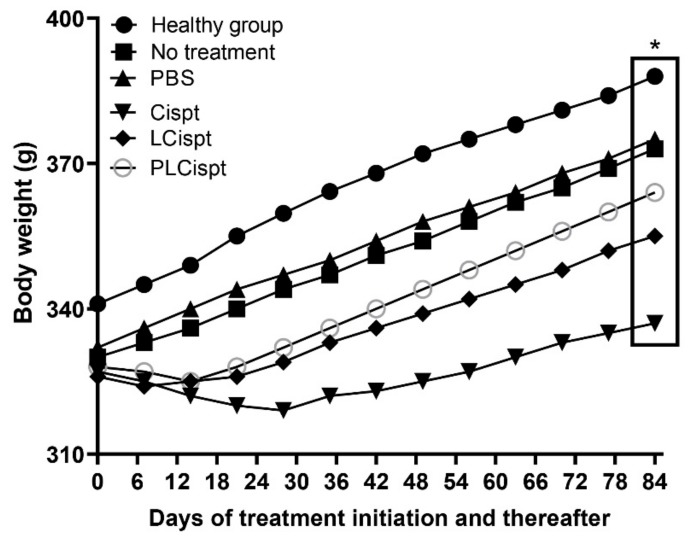
Weight changes in the BC tumor-bearing rats receiving various formulations (control, PBS, Cispt, LCispt, and PLCispt) compared to the healthy control group. The bodyweight gain (BWG) was lower in the standard Cispt rats compared to PLCispt receiver group (3% versus 10%). Statistical analyses were performed using one-way ANOVA and F-tests with * *p* < 0.05. The data are expressed as mean ± SD (*n* = 3).

**Figure 9 ijms-21-00559-f009:**
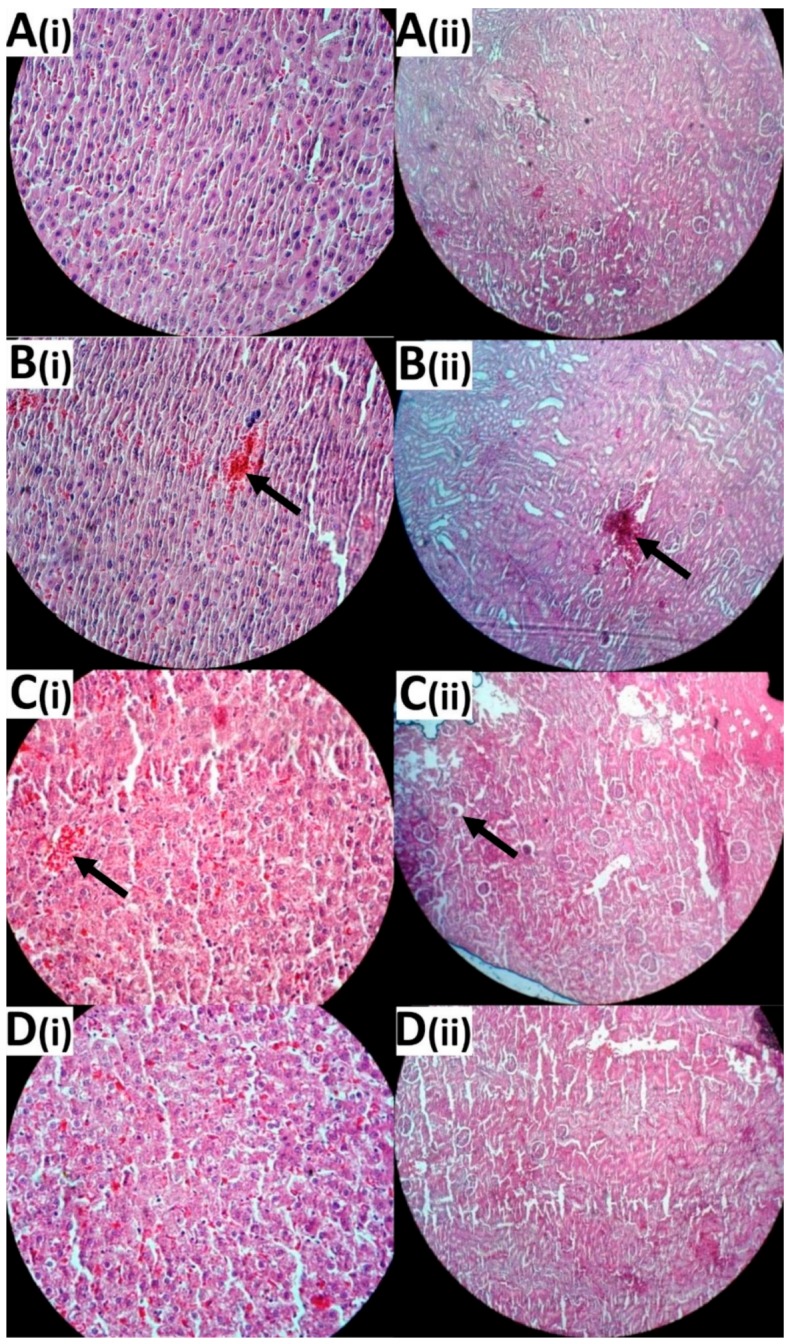
Histological evaluation of (**i**) liver and (**ii**) kidney tissues in (**A**) healthy rats, (**B**) BC-bearing rats receiving Cispt, (**C**) BC-bearing rats receiving LCispt, and (**D**) BC-bearing rats receiving PLCispt. The arrows show cell necrosis. Arrows show the pathological lesions (Magnification ×10).

**Table 1 ijms-21-00559-t001:** The size, size distribution, and zeta potential of liposome, Cispt-loaded liposome (LCispt), and PEGylated liposome (PLCispt).

Batches of Nanoparticles	Size (nm)	Size Distribution	Zeta Potential (mV)	Lipid Composition
Liposome	221.0 ± 11.0	0.40 ± 0.02	−27.0 ± 1.3	Lecithin and cholesterol
LCispt	274.0 ± 12.6	0.058 ± 0.001	−20.0 ± 0.9	Lecithin and cholesterol
PLCispt	251.0 ± 12.0	0.046 ± 0.002	−7.0 ± 0.3	Lecithin and cholesterol

**Table 2 ijms-21-00559-t002:** Size, size distribution, and zeta potential of PLCispt obtained at the production time and two months later.

Batches of Nanoparticles	Size (nm)	Size Distribution	Zeta Potential (mV)
PLCispt (Production day)	251.0 ± 12.0	0.046 ± 0.002	−7.0 ± 0.3
PLCispt (Two months later)	255.2 ± 13.0	0.049 ± 0.002	−6.0 ± 0.3

**Table 3 ijms-21-00559-t003:** Histological evaluation of organ toxicity after treatment with Cispt, LCispt, and PLCispt, and the serum concentrations of blood urea nitrogen (BUN), creatinine, alanine transaminase (ALT), alkaline phosphatase (ALP), and aspartate transaminase (AST) in these groups of animals.

Group	Number of Animals	Organ	Score	BUN (mg/dL)	Creatinine (mg/dL)	ALP (U/L)	ALT (U/L)	AST (U/L)
Cispt	10	Liver	2	67 ± 3.1	3.8 ± 0.2	273± 13.3	284 ± 14.3	365 ± 18.1
		Kidney	1–2					
LCispt	10	Liver	0–1	54 ± 2.6	3.1 ± 0.15	220± 11.1	237± 11.4	290± 14.3
		Kidney	1					
PLCispt	10	Liver	0–1	32 ± 1.5	1.8± 0.09	130± 6.2	138 ± 6.5	170± 8.2
		Kidney	0–1					

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
