# Peer review of "Enhanced Efficacy of PEGylated Liposomal Cisplatin: In Vitro and In Vivo Evaluation"

_ijms, 2020, doi:10.3390/ijms21020559_

Round 1

Reviewer 1 Report

The manuscript “Enhanced efficacy of PEGylated liposomal cisplatin: in vitro and in vivo evaluation” presents a good research, but with a low degree of originality. I hope the authors will improve the introduction and comment more on similar work performed with cisplatin.

The introduction could be improved by a better choice of references. For example, the authors statement “It is a poor prognosis malignancy owing to late detection, diagnosis, and limited treatment choices [2]” is backed up by the reference “Lectins as possible tools for improved urinary bladder cancer management”. I’m sure that there are more direct sources to describe these processes. See also, “Cisplatin (Cispt) is the most commonly used chemotherapeutic agent for the treatment of BC” and its reference LncRNA-MALAT1 mediates cisplatin resistance via miR-101-3p/VEGF-C pathway in bladder cancer. It should be changed with a better source.

Please check all the abbreviations! See for example, loading efficiency (LE) is presented in the materials section. It should appear at the first mention in the text, on row 127. See also BBN and all the manuscript!

Row 131. It is not correct to compare with a totally different drug, in this case, paclitaxel. The authors should explain the low loading efficiencies values.

Row 133-139. This section is not very useful and I also don’t find it correct. Those bands could come from all the other ingredients in the formulation of PEGylated liposomes. The method of FTIR and apparatus are missing.

The authors should comment more on the low release of cisplatin close to only 40% after 48 hours. The low encapsulation efficiency taken together with the low release would mean higher doses of this product (close to 10x the normal cisplatin doses).

Rows 182 to 194. The authors should explain better how the IC50 value was calculated for the Cispt-loaded liposome (LCispt) and PEGylated liposome (PLCispt). If we are talking about μM concentration, how was this concentration calculated? Is the concentration of pure cisplatin that is present in the formultions?

It is strange in my opinion that the IC50 values for LCispt and PLCispt are virtually the same for 24 h and 48 h. A comment on this?

Row 223. Present the values in a table of a figure for clarity.

In figure 7 the data shows 84 weeks. How is this possible? In the first 8 BBN was administered and afterwards the drugs (20 weeks???). Is this correct?

Row 355. Present the approval number and date.

Row 358. Explain what does 1.5 mg of drug means for the loaded formulation. Is the weight of the product administered or the cisplatin net content?

Row 359. It is not very clear what 6 × 72 h means. Please explain in order to avoid confusions.

Author Response

Dear Professor Ms. Heidi Zhao,

We welcome the comments from the reviewers. The manuscript has been revised and comments are addressed below:

Reviewer 1

The manuscript “Enhanced efficacy of PEGylated liposomal cisplatin: in vitro and in vivo evaluation” presents a good research, but with a low degree of originality. I hope the authors will improve the introduction and comment more on similar work performed with cisplatin.

The introduction could be improved by a better choice of references. For example, the authors statement “It is a poor prognosis malignancy owing to late detection, diagnosis, and limited treatment choices [2]” is backed up by the reference “Lectins as possible tools for improved urinary bladder cancer management”. I’m sure that there are more direct sources to describe these processes. See also, “Cisplatin (Cispt) is the most commonly used chemotherapeutic agent for the treatment of BC” and its reference LncRNA-MALAT1 mediates cisplatin resistance via miR-101-3p/VEGF-C pathway in bladder cancer. It should be changed with a better source.

Response: We have strengthened the referencing throughout the manuscript by adding more appropriate references.

Please check all the abbreviations! See for example, loading efficiency (LE) is presented in the materials section. It should appear at the first mention in the text, on row 127. See also BBN and all the manuscript!

Response: We thank the reviewer for noting this error. We have corrected them in the manuscript.

Row 131. It is not correct to compare with a totally different drug, in this case, paclitaxel. The authors should explain the low loading efficiencies values.

Response: We thank to reviewer for his/her supportive comments and advice. It has been corrected in the manuscript (lines 138 and 139).

Row 133-139. This section is not very useful and I also don’t find it correct. Those bands could come from all the other ingredients in the formulation of PEGylated liposomes. The method of FTIR and apparatus are missing.

Response: According to the previous studies which have been mentioned in the manuscript, the bands corresponded to Cispt. Also, the FTIR method has been added in the manuscript (lines 362-367).

The authors should comment more on the low release of cisplatin close to only 40% after 48 hours. The low encapsulation efficiency taken together with the low release would mean higher doses of this product (close to 10x the normal cisplatin doses).

Response: To answer this question, the following line has been added in the manuscript (lines 180 and 181)

Overall, the low amounts of drug release from liposomes, in the present study, reflected the prolonged-release property of these carriers, especially for the PLCispt.

Rows 182 to 194. The authors should explain better how the IC50 value was calculated for the Cispt-loaded liposome (LCispt) and PEGylated liposome (PLCispt). If we are talking about μM concentration, how was this concentration calculated? Is the concentration of pure cisplatin that is present in the formulations?

Response: The IC50 values were calculated using GraphPad Prism software

It is strange in my opinion that the IC50 values for LCispt and PLCispt are virtually the same for 24 h and 48 h. A comment on this?

Response: This low enhancement in the cytotoxicity effects of PLCispt compared to LCispt might be related to the drug release profile of both formulations, in which the amount of released Cispt from PLCispt was slightly lower than LCispt (lines 201-204).

Row 223. Present the values in a table of a figure for clarity.

Response: We thank to reviewer for his/her supportive comments and advice. Please see table figure 7 (line 270).

In figure 7 the data shows 84 weeks. How is this possible? In the first 8 BBN was administered and afterwards the drugs (20 weeks???). Is this correct?

Response: We thank the reviewer for noting this error. The X axis is “days of treatment initiation and thereafter”. It has been corrected in the manuscript.

Row 355. Present the approval number and date.

Response: We thank to reviewer for his/her supportive comments and advice. It has been added in the manuscript (lines 409 and 410).

Row 358. Explain what does 1.5 mg of drug means for the loaded formulation. Is the weight of the product administered or the cisplatin net content?

Response: We thank to reviewer for his/her supportive comments and advice. It is Cispt net content. It has been corrected in the manuscript (line 413)

Row 359. It is not very clear what 6 × 72 h means. Please explain in order to avoid confusions.

Response: We thank to reviewer for his/her supportive comments and advice. It has been corrected in the manuscript (line 413)

Reviewer 2 Report

Ghaferi et al prepared the cisplatin-loaded liposome with or without PEG, and compared their physicochemical characteristics, drug releasing rate, safety and anti-tumor efficacy in vitro and in vivo. However, a lot of reports regarding cisplatin-loaded liposome have been reported so far. Thus, authors should clearly describe the novelty of this study.

Table 1: What is lipid composition of Liposome?

Table 1: What is the reason for the 10 times higher size distribution of Liposome than that od other liposomes (LCispt and PLCispt)

Line 158: figure 7→figure 4

Line 183: figure 4→figure 5

Line 186, 187: INF→IFN

Stability study: Please provide the physicochemical properties (diameter, PDI, zeta-potential) of PLCispt after 2 month storage.

Stability study: Please provide the storage conditions (temperature, humidity, etc).

Line 233-235: Although authors described that PEGylation improves blood retention resulting in enhancing the accumulation in target site, tumor model rats used in this study received all test samples intraperitoneally. This discussion is not adequate. Instead, the effect of PEGylation in abdominal cavity should be discussed.

Line 244: figure 5→figure 7

Figure 7 vertical axis: Weight changes→Body weight

Table 2: In addition to histological evaluation, please provide biochemical parameters reflected hepatic and renal functions such as AST, ALT, BUN and CRE.

Figure 8: Please provide the pictures of liver and kidney in all groups.

Line 353: In introduction, authors described that most case of bladder cancer occur in men. Why did authors use female rats for in vivo evaluation?

Please provide approval number from animal ethics committee.

Author Response

Dear Ms. Heidi Zhao,

We welcome the comments from the reviewers. The manuscript has been revised and comments are addressed below:

Reviewer 2

Ghaferi et al prepared the cisplatin-loaded liposome with or without PEG, and compared their physicochemical characteristics, drug releasing rate, safety and anti-tumor efficacy in vitro and in vivo. However, a lot of reports regarding cisplatin-loaded liposome have been reported so far. Thus, authors should clearly describe the novelty of this study.

Table 1: What is lipid composition of Liposome?

Response: Lecithin and cholesterol. It has been added in the table (line 106 and 107)

Table 1: What is the reason for the 10 times higher size distribution of Liposome than that of other liposomes (LCispt and PLCispt).

Response: The amount of size distribution for free liposome was significantly higher compared to LCispt and PLCispt formulations, which could be resulted from the effects of Cispt as a positive charge molecule in increasing the nanoparticles homogeneity (line 103-105).

Line 158: figure 7→figure 4

Response: We thank the reviewer for noting this error. It has been corrected in the manuscript.

Line 183: figure 4→figure 5

Response: We thank the reviewer for noting this error. It has been corrected in the manuscript.

Line 186, 187: INF→IFN

Response: According to the reference, it is INF.

Stability study: Please provide the physicochemical properties (diameter, PDI, zeta-potential) of PLCispt after 2 month storage.

Response: We thank to reviewer for his/her supportive comments and advice. It has been added as table 2 in the manuscript (line 236 and 237).

Stability study: Please provide the storage conditions (temperature, humidity, etc).

Response: We thank to reviewer for his/her supportive comments and advice. It has been added in the manuscript (line 400 and 401).

Line 233-235: Although authors described that PEGylation improves blood retention resulting in enhancing the accumulation in target site, tumor model rats used in this study received all test samples intraperitoneally. This discussion is not adequate. Instead, the effect of PEGylation in abdominal cavity should be discussed.

Response: To answer this question, the following line has been added in the manuscript (lines 258-261)

 PEGylation in enhancing the potency of liposome nanoparticles after intraperitoneal administration. In this regard, the results of a study showed that PEGylated liposomes could be detected in the blood, 30 h after intraperitoneal injection, while this value for non-PEGylated liposome was 7 h.

Line 244: figure 5→figure 7

Response: We thank the reviewer for noting this error. It has been corrected in the manuscript.

Figure 7 vertical axis: Weight changes→Body weight

Response: We thank to reviewer for his/her supportive comments and advice. It has been added in the manuscript (line 303).

Table 2: In addition to histological evaluation, please provide biochemical parameters reflected hepatic and renal functions such as AST, ALT, BUN and CRE.

Response: These have been added as a table (table 3) in the manuscript (line 314).

Figure 8: Please provide the pictures of liver and kidney in all groups.

Response: They have been added in the manuscript as figure 9 (line 317).

Line 353: In introduction, authors described that most case of bladder cancer occur in men. Why did authors use female rats for in vivo evaluation?

Response: We used female rats according to the previous studies which have been cited in the manuscript (line 241).

Please provide approval number from animal ethics committee.

Response: We thank to reviewer for his/her supportive comments and advice. It has been added in the manuscript (lines 409 and 410).

Round 2

Reviewer 1 Report

The authors tried to improve their paper by correcting some mistakes and in some cases ignored the reviewers comments.

I still think the introduction needs major modifications in order to present the reader some other cisplatin loaded preparation prepared and tested by other authors before. It looks that this team is the first to prepare this kind of formula.

The references are not very good chosen. I strongly advise the authors to go to the direct source. Why so many references are about paclitaxel? Is this taken from an older article?

In some case the reference is not at all related to the subject, but a mean for self-citations. They should be removed! See as example: Moosavi, M.R.; Zare, R. Present Status and the Future Prospects of Microbial Biopesticides in Iran. Agriculturally Important Microorganisms, Springer: 2016; pp. 293-305.

As I said in my first review, I don't consider the FTIR spectra relevant. After the centrifugation and separation the authors isolated cisplatin? If so, why the spectra are not the same?

Author Response

Dear Ms. Heidi Zhao,

We welcome the comments from the reviewers. The manuscript has been revised and comments are addressed below:

Reviewer 1

The authors tried to improve their paper by correcting some mistakes and in some cases ignored the reviewers’ comments.

I still think the introduction needs major modifications in order to present the reader some other cisplatin loaded preparation prepared and tested by other authors before. It looks that this team is the first to prepare this kind of formula.

Response: We have discussed this point (pages 3 and 4) in the “Introduction” (lines 50-60) adding the following line:

“Kates et al. [10] synthesized Cispt-loaded Poly(L-aspartic acid sodium salt) (PAA) nanoparticles (140 ± 4 nm) and evaluated its efficacy on the treatment of BC in vivo environment. The results showed that BC-bearing rats treated with PAA-Cispt nanoparticles had no signs of T1 grade tumors, while 20% of rats treated with Cispt had T1 high grade tumors in the bladder, indicating the role of PAA nanoparticles in increasing the therapeutic effects of Cispt in the treatment of BC. In another study, Sudha et al. [11] synthesized Cispt-loaded nano-diamino-tetrac (NDAT) particles (187 nm) and evaluated its efficacy in the treatment of BC in vivo environment. The results showed that the nanoparticles caused a significant reduction in the tumor volume compared to when Cispt was used (tumor volume of 0.68 and 0.3 cm3 in Cispt and Cispt-loaded NDAT nanoparticles receiver animals, respectively). Liposome nanoparticles are another delivery system, widely used for the treatment of cancer in vitro and in vivo environments [9,12,13].”

The references are not very good chosen. I strongly advise the authors to go to the direct source. Why so many references are about paclitaxel? Is this taken from an older article?

Response: The articles related to Paclitaxel has been cited because of the roles of for example PEG in e.g. reducing drug release and increasing cytotoxicity. We did not compare this drug with cisplatin. So, we prefer to keep these references to support our statements.

In some case the reference is not at all related to the subject, but a mean for self-citations. They should be removed! See as example: Moosavi, M.R.; Zare, R. Present Status and the Future Prospects of Microbial Biopesticides in Iran. Agriculturally Important Microorganisms, Springer: 2016; pp. 293-305.

Response: The mentioned reference by the reviewer is not our paper. As we previously mentioned the reason for putting these references are to support our statements. Also, there is a scientific reason if we cited our published papers; however, we removed this citation.

As I said in my first review, I don't consider the FTIR spectra relevant. After the centrifugation and separation, the authors isolated cisplatin? If so, why the spectra are not the same?

Response: I think there is a confusion. As we discussed in “methods and Materials” section (lines 353-358), we did not isolate cisplatin, but we isolated cisplatin-loaded liposome. The reason for doing FTIR test was to determine whether the chemical structure of cisplatin was preserved after loading into liposomes. So, we performed this test for standard cisplatin and cisplatin-loaded liposome to check if cisplatin structure was changed. Therefore, we believe that this test is completely relevant as many researchers have done this test before.

We hope these answers have addressed your concerns and this article will be accepted

Reviewer 2 Report

None

Author Response

Dear Ms. Heidi Zhao,

Thank you very much.

Regards

Round 3

Reviewer 1 Report

The paper is acceptable in this form.